

# Freshwater macrophytes harbor viruses representing all five major phyla of the RNA viral kingdom *Orthornavirae*

Karyna Rosario[1], Noémi Van Bogaert[1,2], Natalia B. López-Figueroa[1], Haris Paliogiannis[1,3], Mason Kerr[1] and Mya Breitbart[1]

[1] College of Marine Science, University of South Florida, St Petersburg, Florida, United States
[2] Present Address: FVPHouse, Berlare, Belgium
[3] Present Address: MIO-ECSDE, Athens, Greece

## ABSTRACT

Research on aquatic plant viruses is lagging behind that of their terrestrial counterparts. To address this knowledge gap, here we identified viruses associated with freshwater macrophytes, a taxonomically diverse group of aquatic phototrophs that are visible with the naked eye. We surveyed pooled macrophyte samples collected at four spring sites in Florida, USA through next generation sequencing of RNA extracted from purified viral particles. Sequencing efforts resulted in the detection of 156 freshwater macrophyte associated (FMA) viral contigs, 37 of which approximate complete genomes or segments. FMA viral contigs represent putative members from all five major phyla of the RNA viral kingdom *Orthornavirae*. Similar to viral types found in land plants, viral sequences identified in macrophytes were dominated by positive-sense RNA viruses. Over half of the FMA viral contigs were most similar to viruses reported from diverse hosts in aquatic environments, including phototrophs, invertebrates, and fungi. The detection of FMA viruses from orders dominated by plant viruses, namely *Patatavirales* and *Tymovirales*, indicate that members of these orders may thrive in aquatic hosts. PCR assays confirmed the presence of putative FMA plant viruses in asymptomatic vascular plants, indicating that viruses with persistent lifestyles are widespread in macrophytes. The detection of potato virus Y and oat blue dwarf virus in submerged macrophytes suggests that terrestrial plant viruses infect underwater plants and highlights a potential terrestrial-freshwater plant virus continuum. Defining the virome of unexplored macrophytes will improve our understanding of virus evolution in terrestrial and aquatic primary producers and reveal the potential ecological impacts of viral infection in macrophytes.

Corresponding authors
Karyna Rosario,
krosari2@usf.edu
Mya Breitbart, mya@usf.edu

## INTRODUCTION

Over a decade of research has confirmed the critical roles of viruses in the evolution and ecology of wild terrestrial vegetation (*Malmstrom, Melcher & Bosque-Pérez, 2011*; *Roossinck, 2015*; *Shates et al., 2019*). Yet, little is known about plant virus ecology in aquatic systems. The sparse information regarding viruses infecting aquatic vegetation was noted

over 50 years ago when researchers considered viruses as an alternative strategy to control harmful freshwater plants (*Zettler & Freeman, 1972*). This biocontrol idea was based upon evidence from the 1950's demonstrating viral infection in freshwater vascular plants (*MacClement & Richards, 1956*). During the past 70 years there has been little progress in investigating viral infection in aquatic vegetation, with the exception of single-celled phototrophs (*Brussaard, 2004*; *Coy et al., 2018*; *Nagasaki, 2008*; *Van Etten, Agarkova & Dunigan, 2019*), a knowledge gap addressed herein.

In contrast to terrestrial systems where vascular plants dominate, macrophytes, together with phytoplankton, are the principal primary producers in aquatic ecosystems (*Jänes et al., 2017*; *Nõges, Luup & Feldmann, 2010*). Macrophytes are a group of taxonomically diverse photosynthetic organisms that are visible to the naked eye and actively grow, permanently or periodically, in aquatic environments (*Chambers et al., 2008*; *Lesiv, Polishchuk & Antonyak, 2020*). This diverse group includes phototrophs spanning from cyanobacterial mats (Cyanophyta) to seven divisions within Archaeplastida, including macroalgae (Charophyceae, Chlorophyta, Rhodophyta, and Xanthophyta), non-vascular plants (Bryophyta) and vascular plants (Pteridophyta and Spermatophyta) (*Chambers et al., 2008*). Macrophytes play essential roles in aquatic systems by influencing habitat structure and function (*e.g.*, modifying water current and sediment conditions), serving as food sources for a wide range of herbivores, providing habitat and a structurally complex environment, cycling carbon and nutrients, and improving water quality (*Dibble, Thomaz & Padial, 2006*; *Duarte et al., 2013*; *Heck et al., 2008*; *Lamb et al., 2017*; *Lesiv, Polishchuk & Antonyak, 2020*; *Reitsema, Meire & Schoelynck, 2018*; *Rennie & Jackson, 2005*; *Srivastava, Gupta & Chandra, 2008*; *Waycott et al., 2009*). Although macrophytes are important for maintaining aquatic ecosystem health and function, many species require management as they can become harmful when overgrown due to nutrient loading and/or new habitat invasion (*Anderson, 2003*; *Smetacek & Zingone, 2013*). A better understanding of factors shaping macrophyte ecology will lead to more effective conservation and management strategies for aquatic ecosystems (*Chambers et al., 1999*).

Viruses are expected to impact macrophytes given their effects on terrestrial plant ecology and evolution. Studies of plant viral infection in wild populations and those at the interface between agricultural and unmanaged vegetation indicate that viruses play a significant evolutionary role in plants by affecting plant fitness, population dynamics, and diversity (*Kelley et al., 1994*; *Montes, Alonso-Blanco & García-Arenal, 2019*; *Remold, 2002*; *van Mölken & Stuefer, 2011*). Although viruses are often viewed as disease agents, viral infections are prevalent in natural terrestrial systems where viruses often coexist with their plant hosts without causing negative effects, displaying mutualistic or even beneficial interactions (*Boccardo et al., 1987*; *Roossinck, 2011*; *Roossinck, 2015*; *Roossinck & Schultz-Cherry, 2015*). Limited sampling of kelp and seagrass in marine environments indicates that viral infection is also prevalent in marine macrophytes, with over 60% of sampled individuals testing positive for viruses (*McKeown et al., 2018*; *McKeown et al., 2017*; *Van Bogaert et al., 2019*). Similar to what has been observed in terrestrial wild vegetation (*Kamitani et al., 2016*; *Roossinck, Martin & Roumagnac, 2015*; *Susi et al., 2019*), these prevalent macrophyte viral infections are mainly asymptomatic. Research in freshwater

systems has mainly focused on viruses infecting charophytic algae (*Chara* spp.), which are interesting hosts from an evolutionary standpoint due to their ancestral position relative to land plants (*Zhong, Sun & Penny, 2015*). Indeed, *Chara*-infecting viruses have unique features resembling various groups of terrestrial plant viruses and therefore may hold clues regarding plant virus evolution (*Gibbs et al., 2011*; *Vlok, Gibbs & Suttle, 2019*). More virological research on macrophytes will lead to a deeper understanding of their ecology and potential evolutionary links among freshwater, marine, and terrestrial plant viruses.

To broadly investigate freshwater macrophyte associated (FMA) viruses, we surveyed pooled samples of macrophyte species collected at four spring sites in Florida, USA. The springs have significant cultural, ecological, and economic value for the state of Florida and this study joins efforts to investigate understudied viral dynamics in these freshwater systems (*Malki et al., 2020*; *Malki et al., 2021*). The captured diversity includes viruses from all five major phyla of the RNA viral kingdom *Orthornavirae* that are currently recognized by the International Committee on Virus Taxonomy (ICTV). We explored taxonomic affiliations of detected viral sequences to evaluate how FMA viruses fit within the known RNA virosphere (*Koonin et al., 2020*). PCR assays for putative FMA plant viruses demonstrated widespread infections in sampled vascular plants. The detection of two known crop-infecting viral pathogens in submerged macrophytes suggests that terrestrial plant viruses infect underwater plants and highlights a potential terrestrial-freshwater plant virus continuum.

## MATERIALS AND METHODS

### Study site and sample collection

Macrophyte samples were collected during July 2017 from freshwater springs located within four Florida State Parks, namely Ichetucknee, Rainbow, Manatee, and Blue Springs State Parks, in accordance with permit 06011710 from the Florida Department of Environmental Protection. Three sampling points were selected within each spring site (Table 1), starting at the spring head where underground water emerges (Location ID 1) and moving ~0.2 miles downstream in a stepwise manner (Location IDs 2 and 3, respectively). Two samples from each visually distinct macrophyte species observed within each spring location were hand-picked through wading and snorkeling. Each macrophyte sample was rinsed on site with spring water and placed into individual Ziplock© bags. Samples were kept on ice during transport to the lab where collected species were identified based on morphological features whenever possible and stored at −80 °C until further processing.

### Virus particle purification from macrophyte tissues and nucleic acid extraction

Virus particles were purified from macrophyte tissues prior to nucleic acid extractions following previously described methods for submerged aquatic vegetation (*Van Bogaert et al., 2019*). Samples were thawed, rinsed with MilliQ water, and visible epiphytes were carefully removed using sterile scalpel blades. Approximately 200 mg of tissue from each macrophyte species (*i.e.*, 100 mg per each individual plant from a given species from each

**Table 1 Sampling locations and macrophytes collected within four freshwater springs.**

| Spring (ID) | Latitude/Longitude (Location ID) | Collected species (Common name) | Specimen description |
|---|---|---|---|
| Blue | 28.947483/−81.339574 (1) | *Lyngbya wollei** | Mat-forming cyanobacteria |
| | | *Typha spp.* (cattail) | Require perpetually moist soil |
| | | Unidentified | n/a |
| | | *Hydrocotyle umbellate* L. (Pennywort) | Rooted, grows in water or on land |
| | 28.947163/−81.33964 (2) | *Lyngbya wollei* | Mat-forming cyanobacteria |
| | | *Sagittaria kurziana* Glück (Springtape) | Rooted submerged plant |
| | | *Tillandsia usneoides* L. (Spanish moss)** | Epiphytic and rootless 'air-plant' |
| | 28.944765/−81.339414 (3) | *Ludwigia repens* J.R. Forst. (Red ludwigia) | Rooted, grows partially or fully submerged |
| | | *Sagittaria lancifolia* L. (Lanceleaf arrowhead) | Rooted, grows in shallow-water habitats |
| | | *Hydrocotyle umbellata* L. (Pennywort) | Rooted, grows in water or on land |
| | | Unidentifed | n/a |
| | | *Lyngbya wollei** | Mat-forming cyanobacteria |
| Ichetucknee | 29.984065/−82.761744 (1) | Unidentified | n/a |
| | | *Vallisneria americana* Michx. (Tapegrass) | Rooted submerged plant |
| | | *Ludwigia repens* J.R. Forst. (Red ludwigia) | Rooted, grows partially or fully submerged |
| | | *Hydrocotyle umbellata* L. (Pennywort) | Rooted, grows in water or on land |
| | | *Chara sp.* (Muskgrass) | Branched macroalgae |
| | 29.982173/−82.760423 (2) | *Vallisneria americana* Michx. (Tapegrass) | Rooted submerged plant |
| | | *Hydrocotyle umbellata* L. (Pennywort) | Rooted, grows in water or on land |
| | 29.981734/−82.760234 (3) | *Ludwigia repens* J.R. Forst. (Red ludwigia) | Rooted, grows partially or fully submerged |
| | | *Ceratophyllum demersum* L. (Hornwort) | Grows free-floating and submerged |
| | | *Hydrocotyle umbellata* L. (Pennywort) | Rooted, grows in water or on land |
| | | *Vallisneria americana* Michx. (Tapegrass) | Rooted submerged plant |
| Manatee | 29.489562/−82.977069 (1) | *Lyngbya wollei** | Mat-forming cyanobacteria |
| | 29.489403/−82.977678 (2) | *Lyngbya wollei** | Mat-forming cyanobacteria |
| | 29.489216/−82.978378 (3) | *Lyngbya wollei** | Mat-forming cyanobacteria |
| | | Unidentified | n/a |
| | | Unidentified | n/a |
| | | Unidentified | n/a |
| Rainbow | 29.1023/−82.437633 (1) | *Hydrilla verticillata* (L. f.) Royle (Waterthyme) | Rooted submerged plant |
| | | *Potamogeton pectinatus* L. (Fennel pondweed) | Rooted submerged plant |
| | | *Utricularia* sp. (Bladderwort) | Submerged or free-floating carnivorous plant |
| | | *Myriophyllum heterophyllum* Michx. (Broadleaf watermilfoil) | Rooted submerged plant |
| | | *Potamogeton illinoensis* Morong (Illinois pondweed) | Rooted submerged plant |
| | | *Sagittaria kurziana* Glück (Springtape) | Rooted submerged plant |

| Spring (ID) | Latitude/Longitude (Location ID) | Collected species (Common name) | Specimen description |
|---|---|---|---|
| | 29.101762/−82.437174 (2) | *Myriophyllum heterophyllum* Michx. (Broadleaf watermilfoil) | Rooted submerged plant |
| | | *Potamogeton illinoensis* Morong (Illinois pondweed) | Rooted submerged plant |
| | | *Sagittaria kurziana* Glück (Springtape) | Rooted submerged plant |
| | | *Utricularia* sp. (Bladderwort) | Submerged or free-floating carnivorous plant |
| | | *Ceratophyllum demersum* L. (Hornwort) | Grows free-floating and submerged |
| | | *Najas guadalupensis* (Spreng.) Magnus (Southern waternymph) | Rooted submerged plant |
| | 29.101305/−82.436856 (3) | *Sagittaria kurziana* Glück (Springtape) | Rooted submerged plant |
| | | *Cladium jamaicense* L. (Saw-grass) | Rooted, water-loving (grows in wet or dry soil) |
| | | *Ludwigia repens* J.R. Forst. (Red ludwigia) | Rooted, grows partially or fully submerged |
| | | *Hydrocotyle umbellata* L. (Pennywort) | Rooted, grows in water or on land |

**Notes:**
[*] *Lyngbya wollei* is currently regarded as a synonym of *Microseira wollei*.
[**] Spanish moss is not considered a macrophyte.
n/a, not available.

location) were placed in a 1.5 ml Zymo© bead beating tube containing 2 mm ceramic beads (Zymo Research, Irvine, CA, USA) with 800 μl of Suspension Medium (SM) buffer (0.1 M NaCl, 50 mM Tris-HCl (pH 7.5), 10 mM $MgSO_4$). Tissues were homogenized through bead-beating using a Fisherbrand™ Bead Mill 4 Homogenizer (Fisher Scientific, Waltham, MA, USA) at maximum speed for 90 s. Homogenates were then centrifuged at 4 °C for 10 min at 10,000×g and supernatants were filtered through a 0.22 μm Sterivex (Millipore, Burlington, MA, USA). Filtrates containing partially purified virus particles were treated with chloroform (20% final concentration), vortexed vigorously, and incubated at room temperature for 10 min to disrupt any remaining lipid-containing entities such as cell membranes. After centrifuging the chloroform mixture for 30 s at room temperature, the aqueous fraction was collected and nuclease-treated to remove non-encapsidated nucleic acids. Nuclease treatment was performed by incubating the aqueous fraction for 1.5 h at 37 °C with a nuclease cocktail consisting of 1X Turbo DNase Buffer (Invitrogen, Waltham, MA, USA), 21 U of Turbo DNase (Invitrogen, Waltham, MA, USA), 4.5 U of Baseline-ZERO DNase (Epicentre, Paris, France), 112.5 U Benzonase (EMD Millipore, Burlington, MA, USA), and 400 U RNase I (Ambion, Austin, TX, USA) (*Ng et al., 2012*; *Victoria et al., 2009*). Nucleases were inactivated with 20 mM EDTA (pH = 8.0) prior to nucleic acid extractions. Nucleic acids were extracted from 200 μl of purified viral fraction in one of two ways (Table 2). Samples that were used for an initial round of next-generation sequencing (NGS) were extracted using the QIAamp MinElute Virus Spin kit (Qiagen, Hilden, Germany). Samples used for a second NGS run were extracted using the RNeasy kit (Qiagen, Hilden, Germany) with the on-column DNase I digestion. Both types of extractions were performed following manufacturer's protocols.

**Table 2** Methods used for preparing and sequencing samples through two independent NGS runs.

| Site | NGS round | Extraction kit (Qiagen) | Reverse transcription approach* | Product clean-up** | Library type | # of indexing PCR cycles | Illumina platform |
|------|-----------|--------------------------|-------------------------------|---------------------|--------------|--------------------------|-------------------|
| Blue | 1 | QIAamp MinElute Virus Spin kit | Random hexamers | AMpure XP beads | cDNA | 20 | NextSeq |
| | 2 | RNeasy kit | Random hexamers | Spin Column | cDNA | 18 | HiSeq |
| | 2 | RNeasy kit | SISPA | Spin Column | SISPA | 5 | HiSeq |
| Ichetucknee | 1 | QIAamp MinElute Virus Spin kit | Random hexamers | AMpure XP beads | cDNA | 20 | NextSeq |
| | 2 | RNeasy kit | Random hexamers | Spin Column | cDNA | 18 | HiSeq |
| | 2 | RNeasy kit | SISPA | Spin Column | SISPA | 5 | HiSeq |
| Manatee | 1 | QIAamp MinElute Virus Spin kit | Random hexamers | AMpure XP beads | cDNA | 20 | NextSeq |
| | 2 | RNeasy kit | Random hexamers | Spin Column | cDNA | 18 | HiSeq |
| | 2 | RNeasy kit | SISPA | Spin Column | SISPA | 5 | HiSeq |
| Rainbow | 1 | QIAamp MinElute Virus Spin kit | Random hexamers | AMpure XP beads | cDNA | 20 | NextSeq |

**Notes:**
* Distinguishes if cDNA was obtained using random hexamers included with the Superscript First-Strand Synthesis System (Invitrogen) or primers used for sequence-independent, single-primer amplification (SISPA).
** Procedure used to clean-up Klenow reaction products (cDNA libraries) and SISPA products prior to NGS library preparation. The spin column procedure was performed with the DNA Clean & Concentrator®-25 kit (Zymo Research).

## Reverse-transcription for NGS library preparation

Extracted nucleic acids from individual macrophyte species were pooled by spring site, resulting in the following four pools: Blue (12 species), Ichetucknee (11 species), Manatee (six species), and Rainbow (16 species) (Table 1). Pooled nucleic acids were reversed-transcribed for two independent NGS efforts (Table 2). Reverse transcription was performed with the Superscript IV First-Strand Synthesis System for RT-PCR (Invitrogen, Waltham, MA, USA) using random hexamers provided by the manufacturer or a random primer tagged with a known linker sequence following manufacturer's protocols. Products from the former were used without pre-amplification for NGS library preparation (cDNA libraries), whereas products from the latter were used for sequence-independent, single-primer amplification (SISPA libraries, see below) (Table 2). Reverse-transcribed products obtained with random hexamers for cDNA libraries were subjected to second-strand synthesis with the Klenow Fragment DNA Polymerase (New England Biolabs, Ipswich, MA, USA). For cDNA libraries, 80 μl of double-stranded reverse-transcribed product were prepared from each spring site pool and purified with either Agencourt AMpure XP beads (Beckman-Coulter, Brea, CA, USA) or the DNA Clean & Concentrator®-25 (Zymo Research, Irvine, CA, USA) for fragmentation and NGS library preparation (Table 2).

## SISPA for NGS library preparation

Randomly generated cDNA products used for SISPA were obtained using Primer_A (5′-GTTTCCCAGTCACGATANNNNNNNNN-3′) (Gaynor et al., 2007) and Primer_N1-8 (5′-CCTTGAAGGCGGACTGTGAGNNNNNNNNN-3′) (Ng et al., 2015) in

separate reactions. Complementary strands for these reverse-transcribed products were synthesized with the Klenow Fragment DNA Polymerase (New England Biolabs, Ipswich, MA, USA). Double-stranded cDNA products were then amplified using 2 μM of the appropriate primer containing the linker sequence alone (underlined primer sequence above) (Table S1). The PCR reaction contained 5 μl of template, 3.7 U AmpliTaq Gold polymerase (Thermo Fisher Scientific, Waltham, MA, USA), 4 mM MgCl$_2$, 0.25 mM dNTPs, and 1X PCR Gold buffer in a 50 μl reaction volume. For reactions using Primer_N1, thermocycling conditions were performed with an initial denaturation at 95 °C for 5 min, followed by 5 cycles of [95 °C for 1 min, 59 °C for 1 min, 72 °C for 90 s], 40 cycles of [95 °C for 30 s, 59 °C for 30 s, 72 °C for 90 s with an increased extension time of 2 s per cycle], and a final extension at 72 °C for 10 min. Reactions with Primer_B used the same thermocycling settings with the exception of no incremental extension time as the 40 cycles progressed. SISPA products were confirmed through gel electrophoresis using a 1.5% agarose gel stained with ethidium bromide. After visualization, all SISPA products were cleaned with the Zymo DNA & Concentrator Kit-25 (Zymo Research, Irvine, CA, USA). SISPA products from Primer A/B and N1-8/N reactions were pooled by spring site by combining 40 μl of cleaned product from each reaction prior to NGS library preparation.

## NGS library preparation

Samples were sequenced through two independent NGS runs of opportunity (Table 2). Non-amplified double-stranded cDNA samples were fragmented to 300 bp using a Covaris M220 instrument and used as templates for NGS library preparation (cDNA libraries), whereas SISPA products were not further fragmented prior to library preparation (SISPA libraries). All libraries were prepared for multiplexing using the Accel-NGS 1S Plus DNA Library kit for Illumina Platforms (Swift Biosciences, Ann Arbor, MI, USA) following the manufacturer's instructions for DNA inputs <1 ng/μl for the cDNA libraries and >1 ng/μl for SISPA libraries. For the first NGS round, four cDNA libraries representing pooled samples from each spring site were paired-end sequenced (2 × 150 bp) using a mid-output v2.5 (300 cycles) kit on a NextSeq 500 platform (Illumina, San Diego, CA, USA) at the University of Colorado BioFrontiers Next-Gen Sequencing core facility. The second NGS round included cDNA ($n = 3$) and SISPA ($n = 3$) libraries from the Blue, Ichetucknee, and Manatee spring sites. Libraries from the second round were commercially paired-end sequenced (2 × 150 bp) on a HiSeq system (Illumina, San Diego, CA, USA) at GENEWIZ. Raw NGS data can be found in the Sequence Read Archive (SRA) database under BioProject accession number PRJNA826216.

## Sequence analysis

Sequences from both NGS rounds were analyzed using the University of South Florida high performance computing cluster. Raw sequences were trimmed for quality and to remove indexing adapters and SISPA primers (if applicable) using Trimmomatic version 0.36.0 (*Bolger, Lohse & Usadel, 2014*) with default parameters except for a read head crop of 10 bp instead of zero. Sequence quality after trimming was verified with FastQC version

0.11.5 (*Andrews, 2010*). Various assembly strategies using the SPAdes assembly toolkit (*Nurk et al., 2013*), including command line flags for rna-, single-cell, and meta-SPAdes, were evaluated and final strategies were selected based on the approach producing the longest contigs as determined by QUAST (*Gurevich et al., 2013*). Due to the high number of indexing PCR cycles (Table 2), quality-filtered sequences from cDNA libraries were assembled following a pipeline for PCR amplified libraries (*Roux et al., 2019*). To do this, identical reads (no mismatches) were deduplicated using the Clumpify tool (parameters: "dedupe subs=0 passes=2") from the BBtools package (sourceforge.net/projects/bbmap/). Deduplicated sequences were then assembled using single-cell SPAdes (*Bankevich et al., 2012*). Quality-filtered sequences from SISPA libraries were assembled using metaSPAdes v 3.11.1 with default parameters (*Nurk et al., 2017*). For all libraries, assembled contigs larger than 1,000 bp were selected using the Galaxy public server (usegalaxy.org) (*Afgan et al., 2018*) and compared against the NCBI Reference Sequence viral protein database (RefSeq Release number 93, https://www.ncbi.nlm.nih.gov/refseq/) using BLASTx (e-value $< 10^{-10}$). BLAST results were explored using MEGAN6 Community Edition (*Huson et al., 2016*) to identify contigs with matches to RNA viral sequences. These contigs were then compared against the GenBank non-redundant (nr) database (BLASTx, e-value 0.001) to remove contig sequences that had higher identities with non-viral sequences (*i.e.*, false positives). Libraries from NGS Round 2 were accidentally contaminated with turtlegrass virus X (TVX; accession number MH077559). Therefore, contigs representing TVX were also removed from further analyses.

Once viral contigs were identified from each library, a non-redundant file containing contigs from all libraries was created for read mapping analyses. To do this, open reading frames (ORFs) >450 nt were identified using the *getorf* application from the EMBOSS suite (*Rice, Longden & Bleasby, 2000*) as implemented in the Galaxy public server. These ORFs were then used to identify viral contigs containing ORFs sharing >90% identity using CD-HIT (*Fu et al., 2012*). Trimmed and deduplicated forward reads from each library were mapped to the non-redundant viral contig file using BowtieBatch v 1.0.1 and Read2RefMapper v 1.0.1 applications in the CyVerse Cyberinfrastructure (*Goff et al., 2011*) as part of the iVirus pipeline (*Bolduc et al., 2017*). Reads were mapped if they shared >90% identity with a given contig and contigs were considered present in a given sample pool if reads mapped to >75% of the contig length. The number of reads mapping to a given contig was normalized by contig length. Read mapping was used to evaluate the relative distribution of contigs within a given sample pool and whether contigs were present in more than one pool (*i.e.*, spring site). The relative distribution of viral contigs in each pool was summarized in a heatmap created using the *superheat* package in R (https://rlbarter. github.io/superheat/).

## Near-complete genomes

Viral contigs approximating complete genomes or genome segments based on known lengths for their predicted taxonomic group were annotated using Geneious v R8. Potential assembly errors in some contigs, based on known genome organizations, were verified by reassembling reads and contigs using the default Geneious overlap layout consensus

assembler and/or mapping against the original contig using default parameters to evaluate coverage across the genome. ORFs were compared against GenBank nr and Conserved Domain (CDD) databases for annotation purposes. If no significant matches were found in these databases, ORFs were compared against potential remote homologs using HHpred (*Hildebrand et al., 2009*) as implemented in the MPI Bioinformatics Toolkit public server (https://toolkit.tuebingen.mpg.de/) (*Zimmermann et al., 2018*). Near-complete RNA viral genomes or segments are available through GenBank under accession numbers ON125107 to ON125143.

## Phylogenetic analyses

Novel RNA viral sequences identified in this study were compared to previously reported RNA viruses by constructing RNA-directed RNA polymerase (RdRp) phylogenetic trees to evaluate taxonomic affiliations. Only FMA RdRp amino acid sequences that did not contain early stop codons and were >60% the expected length based on reported sequences for a given taxon were included in the analysis. To construct phylogenetic trees, the most similar RdRp amino acid sequences to a given sequence of interest were retrieved through BLAST. Additionally, representative curated RdRp amino acid sequences previously used for a comprehensive analysis of the five major branches of the global RNA virome were retrieved from the supplemental materials provided by *Wolf et al. (2018)*. For each group of interest, sequences were aligned using the MAFFT alignment server (*Katoh, Rozewicki & Yamada, 2019*). Homologous protein sequences were automatically added based on structural alignments from the Database of Aligned Structural Homologs (DASH) to guide alignments, but these sequences were removed from the output alignment file. Poorly aligned regions were removed from alignments using TrimAl (*Capella-Gutierrez, Silla-Martinez & Gabaldon, 2009*) with the *gappyout* method as implemented in the Phylemon2 server (*Sanchez et al., 2011*). The trimmed alignments were used to construct maximum likelihood (ML) phylogenetic trees using PhyML (*Guindon et al., 2010*) with default parameters and automatic selection of best substitution model based on Akaike information criterion (*Lefort, Longueville & Gascuel, 2017*). Support for specific nodes on the trees were assessed using the approximate likelihood ratio test (aLRT) with the nonparametric Shimodaira-Hasegawa-like procedure (*Guindon et al., 2010*). Output tree files were visualized and edited using the *ggtree* R package (*Yu, 2020*; *Yu et al., 2017*).

## PCR assays for detecting putative plant viruses

Since NGS was performed on pooled macrophytes from each site, PCR assays were designed to determine which macrophyte species contained putative plant viruses. Specific PCR primers for each putative FMA plant virus were designed using Primer3 (*Untergasser et al., 2012*) (Table S1) and applied to nucleic acid extracts from purified virus particles from individual macrophyte species collected at each spring site. Single-stranded cDNA was synthesized from extracts using the Superscript IV First-Strand Synthesis System for RT-PCR (Invitrogen, Waltham, MA, USA) with random hexamers provided by the manufacturer. All PCRs were performed using the AmpliTaq Gold™ 360 Master Mix with GC enhancer (Thermo Fisher Scientific, Waltham, MA, USA). Each 25 μl PCR reaction

contained 2 μl of cDNA, 1X AmpliTaq Gold™ 360 Master Mix, 0.96 μM of each primer, and 1 μl 360 GC Enhancer. Thermocycling conditions were performed with an initial denaturation at 95 °C for 10 min, followed by 40 cycles of [95 °C for 30 s, 50 °C for 30 s, 72 °C for 60 s], and a final extension at 72 °C for 10 min. A sample was considered positive for a given virus if a single band of the expected length (Table S1) was observed through gel electrophoresis using a 1.5% agarose gel stained with ethidium bromide.

# RESULTS AND DISCUSSION

## FMA viruses are diverse and dominated by positive-stranded RNA viruses

This study surveyed the diversity of RNA viruses associated with macrophytes in four freshwater springs located in Florida, USA. Viromic sequencing efforts resulted in the detection of 156 distinct RNA viral contigs >1 kilobase (kb) in length, 37 of which represent near-complete genomes or genomic segments (Data S2, Table S2). Two distinct approaches using two independent NGS runs of opportunity were used to sequence viral nucleic acids purified from macrophyte tissues with the goal of capturing a diversity of RNA viruses (Table 2). One approach targeted viral cDNA without preamplification for NGS (cDNA libraries). The second approach exploited SISPA, a random amplification technique prior to library preparation that has been previously used for viral discovery in macrophytes (*Van Bogaert et al., 2019*). The 156 FMA viral contigs presented here describe the combined results, although it is noteworthy that each NGS round and approach identified unique viral contigs (Fig. S1). When comparing results from the same sequencing round (NGS Round 2), it is clear that few viral contigs were recovered from the SISPA libraries that were not also identified in the cDNA libraries. Therefore, cDNA libraries were a more fruitful approach for viral discovery in freshwater macrophytes (Fig. S1).

Viral contigs were detected in pooled macrophyte samples from each of the four surveyed spring sites, including Blue ($n = 49$), Ichetucknee ($n = 76$), Manatee ($n = 30$), and Rainbow ($n = 16$) (Fig. S2). Identified contigs in each spring site were dominated by those representing positive-sense, single-stranded (+) RNA viruses (Fig. 1A). This is consistent with the dominance of +RNA viral types identified in land plant (*Dolja, Krupovic & Koonin, 2020*) and global RNA viromes (*Wolf et al., 2018*), including those from aquatic environments (*Vlok, Lang & Suttle, 2019b; Wolf et al., 2020*). Notably, over 50% of the contigs from macrophyte pools from all the investigated springs were most similar to viral sequences retrieved from aquatic organisms or environments (Fig. 1B).

FMA viral contigs spanned five major phyla within the *Orthornavirae* kingdom (RdRp-encoding viruses) as well as unclassified viruses that have not been accommodated within the current taxonomic framework accepted by the ICTV (*Koonin et al., 2020*) (Fig. 1A). Due to low amino acid sequence identities to known viruses (as low as 20% in some cases, Table S2), most genomic sequences representing novel viruses were labelled here at the order rank to evaluate trends at broader levels of resolution. Datasets from all spring sites were dominated by contigs most similar to members of the order *Picornavirales* and

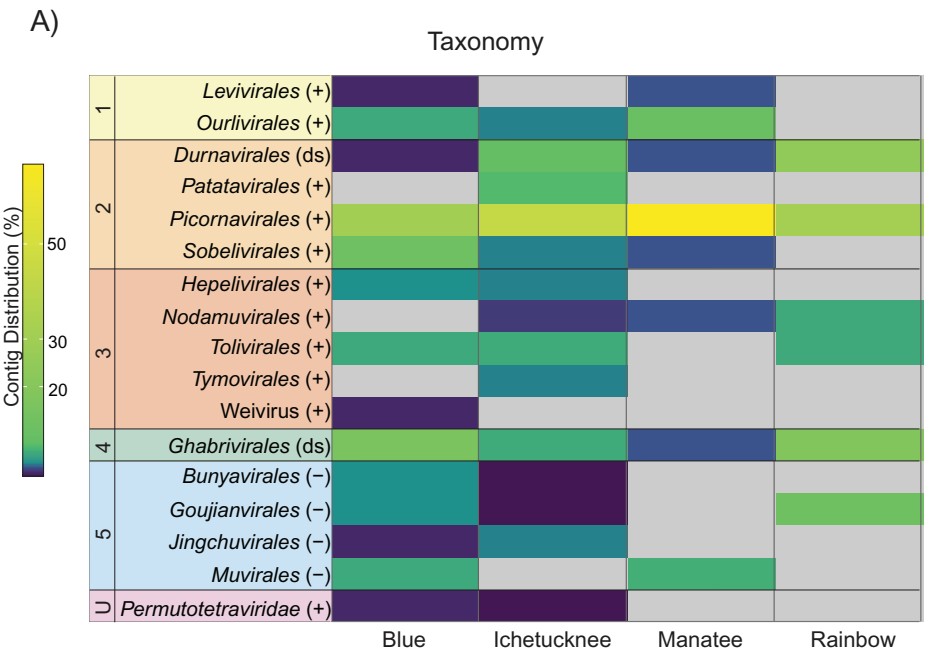

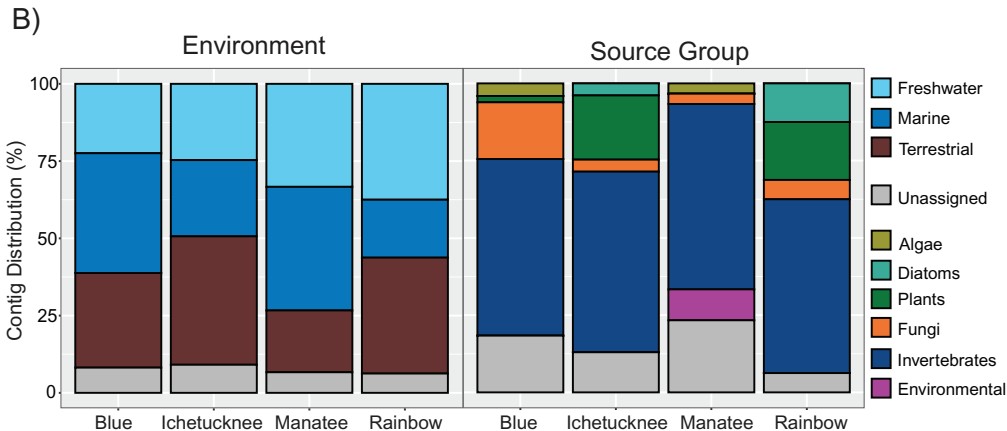

**Figure 1 Distribution of freshwater macrophyte associated (FMA) viral contigs.** (A) Heatmap showing contig distribution based on taxonomic groups. (B) Graphs showing contig distribution based on information about the closest BLASTx match, including type of environment (left) and isolation source organism (right). The color scale on the heatmap represents low (dark purple) to high (yellow) proportion of contigs in a given taxonomic group based on the total number of viral contigs identified in a given spring dataset. Gray color on the heatmap indicates taxonomic groups that were not detected in a spring dataset. Numbers on the left-hand side of the heatmap highlight groups representing each of the five major phyla of the *Orthornavirae* kingdom, including *Lenarviricota* (1), *Pisuviricota* (2), *Kitrina-viricota* (3), *Duplornaviricota* (4), and *Negarnaviricota* (5), whereas the letter "U" specifies an unclassified group. Genome types for each group are indicated within parentheses, including positive (+) and negative (−) single-stranded and double-stranded (ds) RNA viruses. 'Unassigned' in bottom panels refers to contigs that could not be assigned to a given category due to BLASTx matches to viruses from multiple categories.

contained contigs representing members of the orders *Durnavirales* and *Ghabrivirales* (Fig. 1A). However, few contigs were shared across macrophyte pools (Fig. S2). This was expected considering that pools were composed of different macrophytes with little species

overlap and no single macrophyte species was collected from all four spring sites (Table 1). The highest number of shared contigs (*n* = 6) was observed between Manatee and Blue datasets (Table S2). Manatee had the lowest diversity of collected macrophytes, but half of the macrophytes represented the cyanobacterium *Lyngbya wollei*. Blue was the only other spring where *L. wollei* was found, and it was collected from each of the three collection locations within the spring. However, it remains to be determined if shared contigs from Blue and Manatee datasets represent viruses associated with *L. wollei*.

## FMA viruses are most similar to viruses infecting a diversity of hosts

Although the survey presented here was conducted in an attempt to identify potential macrophyte-infecting viruses, identified FMA contigs shared similarities with viruses infecting diverse hosts. More than half of the identified contigs in each pool were most similar to viruses associated with invertebrates (Fig. 1B). All of the datasets contained contigs most similar to mycoviruses (*i.e.*, fungi-infecting viruses) and viruses infecting photosynthetic organisms. Putative plant viruses were identified in macrophyte pools from three spring sites, including Blue, Ichetucknee, and Rainbow. Viruses similar to diatom-infecting viruses were identified in Ichetucknee and Rainbow, whereas viruses most similar to those infecting algae were identified in Blue and Manatee macrophyte pools.

Given the low identities to known viruses it is not possible to predict the hosts of most FMA viruses identified here based on sequence information alone. This is further complicated by shared evolutionary histories among plant-, invertebrate- (*i.e.*, arthropods), and fungi-infecting viruses that result from complex symbiotic relationships among their hosts (*Dolja & Koonin, 2018*; *Dolja, Krupovic & Koonin, 2020*; *Lefeuvre et al., 2019*; *Roossinck, 2019*). Symbiotic associations may allow extensive horizontal virus transfers among disparate hosts. For example, the closest relatives of plant viruses include fungal and arthropod viruses, reflecting strong ties among their hosts (*Dolja, Krupovic & Koonin, 2020*; *Lefeuvre et al., 2019*; *Roossinck, 2019*). Moreover, some plant viruses may replicate within fungi (*Andika et al., 2017*; *Mascia et al., 2019*) and arthropod vectors (*Hogenhout et al., 2008*) further highlighting cross-kingdom viral infections and blurring the lines of what is called a 'plant virus' (*Lefeuvre et al., 2019*).

Despite our limited ability to predict hosts, the viral diversity captured here indicates that macrophytes harbor novel viruses infecting these essential primary producers and members of their holobionts. Macrophytes may also serve as reservoirs for viruses infecting organisms that interact with these autotrophs (*e.g.*, invertebrates). Below we outline taxonomic relationships for each phylum representing the five major branches of the *Orthornavirae* (*Koonin et al., 2020*) to highlight how the viral diversity recovered from freshwater macrophytes fits within the known RNA virosphere. Phylogenetic relationships did not tease apart potential hosts in many cases due to overlapping viromes among fungi, plants, and metazoans (*Dolja & Koonin, 2018*; *Dolja, Krupovic & Koonin, 2020*) and similarities to viruses for which definitive hosts have not been identified. Nevertheless, the genetic data presented here provides a critical starting point to design targeted assays to further investigate FMA viruses and their hosts.

**Branch 1: *Lenarviricota* FMA viruses include putative bacterial and invertebrate viruses**

The phylum *Lenarviricota* occupies the most basal position of the *Orthornavirae* RdRp tree and includes viruses that infect prokaryotes and eukaryotes (*Callanan et al., 2021*; *Koonin et al., 2020*; *Wolf et al., 2018*). *Lenarviricota* FMA viruses include putative members of the class *Leviviricetes* and order *Ourlivirales*, designated here FMA levi-like viruses and FMA ourli-like viruses, respectively.

The majority of *Lenarviricota* FMA viral contigs (8 out of 10) are most similar to members of the *Ourlivirales*, including one near-complete genome (Fig. 2, Table S2). Currently classified members of this order fall within the family *Botourmiaviridae* and include plant viruses and mycoviruses (*Ayllón et al., 2020*). However, a diversity of invertebrate-associated viruses originally labelled as 'narna-like' viruses also belong to this family (*Shi et al., 2016*; *Wolf et al., 2018*). Although invertebrate-associated botourmiavirus hosts remain unverified, intermediate to high (>0.1% to >1% of non-ribosomal RNA reads) abundance levels for several of these viruses in sampled specimens suggest that they are transcriptionally active in invertebrates (*Shi et al., 2016*). All FMA ourli-like viral sequences are most similar to invertebrate-associated botourmiaviruses from aquatic environments, with the exception of FMA ourli-like virus 4 which is most similar to a virus discovered from the root of an apple tree (apple narna-like virus 2) (Table S2). Phylogenetic analyses confirmed that FMA ourli-like viral sequences cluster within the *Botourmiaviridae* clade (Fig. S3).

Botourmiaviruses infecting different host groups have distinguishing genome features. Plant-infecting botourmiviruses (genus *Ourmiavirus*) are segmented, with each segment encoding a single protein including RdRp, movement and capsid proteins. On the other hand, mycovirus genomes belonging to the *Botoulivirus*, *Magoulivirus* and *Scleroulivirus* genera are monocistronic and non-segmented, only encoding for the RdRp. Reported invertebrate-associated botourmiaviruses are non-segmented and encode RdRp alone, similar to mycoviruses from this group, or exhibit a novel organization encoding RdRp and capsid proteins in a putative dicistronic genome (*Shi et al., 2016*). Moreover, several of the invertebrate-associated monocistronic botourmiaviruses contain a picorna-like helicase domain (pfam 00910) within the RdRp ORF that has not been observed in other members of the *Ourlivirales*. One such virus is Wenzhou shrimp virus 10, which was presumed to be transcriptionally active in sampled shrimp based on a high proportion of viral RNA transcripts (*Shi et al., 2016*). The FMA ourli-like virus 1 near-complete genome length (~4.2 kb) and organization matches that of Wenzhou shrimp virus 10 and other monocistronic invertebrate-associated botourmiaviruses (Fig. 2). The remaining FMA viral sequences were most similar to monocistronic and bicistronic invertebrate-associated botourmiaviruses and the monocistronic Apple narna-like virus 2 (Table S2). We were not able to distinguish or predict genome organizations based on the phylogenetic position of individual invertebrate-associated botourmiaviruses. Nevertheless, similarities to viruses containing genomic features that have only been observed in invertebrate-associated

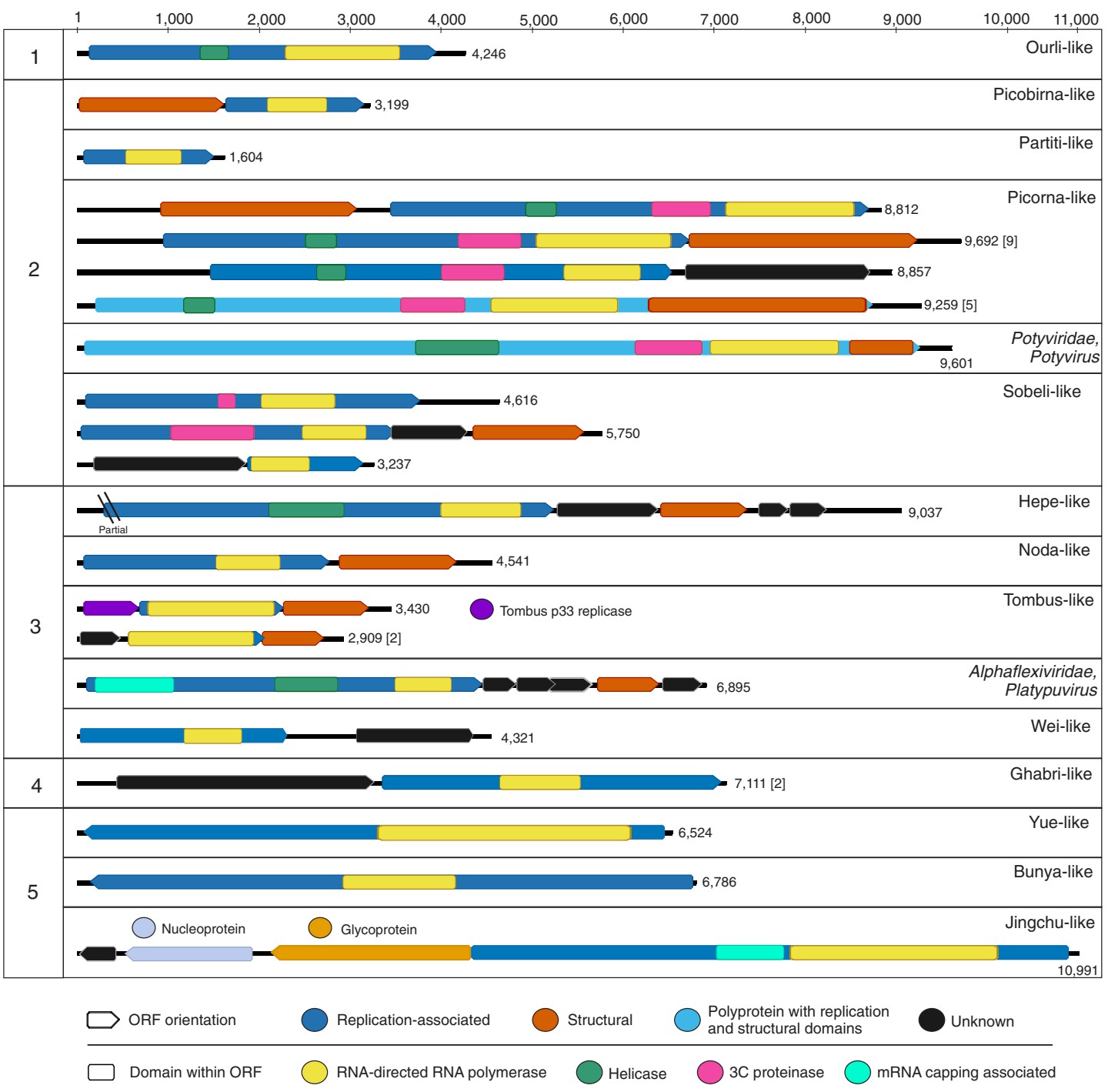

**Figure 2  Genome schematics of near-complete freshwater macrophyte associated (FMA) viral genomes or segments.** Numbers across the top depict a genome size ruler, whereas numbers on the left-hand side group sequences based on the five major branches of the *Orthornavirae* kingdom representing the *Lenarviricota* (1), *Pisuviricota* (2), *Kitrinaviricota* (3), *Duplornaviricota* (4), and *Negarnaviricota* (5) phyla. Numbers on the right-hand side of each genome schematic specify genome size. Genomes with a similar organization are represented by a single schematic and the total number of represented genomes is summarized within brackets. The legend at the bottom of the figure highlights open reading frames (ORF) and predicted protein domains observed in more than one sequence. ORFs observed in a single genome or segment are indicated within their respective panels.

viruses indicate that FMA ourli-like viruses may infect invertebrates rather than fungi or macrophytes.

The remaining two *Lenarviricota* FMA viral contigs (FMA levi-like virus 1 and 2) are most similar to a virus identified from a freshwater mollusk, namely Wenzhou levi-like virus 1 (Table S2). Note that levi-like viruses are distinct from members of the family *Fiersviridae* (formerly named *Leviviridae*), which infect gram-negative bacteria (*Bollback & Huelsenbeck, 2001*). Although levi-like viruses are expected to infect prokaryotes, these viruses have only been discovered through metaviromics and metatranscriptomic studies investigating viruses associated with invertebrates (*Shi et al., 2016*) and environmental samples (*Callanan et al., 2020*; *Krishnamurthy et al., 2016*; *Starr et al., 2019*). Phylogenetic analyses confirmed that FMA levi-like RdRp sequences cluster with novel levi-like viruses (Fig. S3). Specifically, FMA levi-like sequences group in a small clade with levi-like viruses associated with freshwater mollusks and crustaceans. Interestingly, FMA levi-like viruses were detected in the Manatee and Blue datasets, which were the only macrophyte pools that included extracts from *L. wollei* cyanobacterial mats. PCR data confirmed the association of FMA levi-like viral sequences with *L. wollei* (Table 3). Future work is needed to definitively determine if FMA levi-like viruses infect *L. wollei*. This is a topic of substantial interest since the presence of cyanobacterial mats in freshwater systems signals ecosystem degradation (*Hudon, Sève & Cattaneo, 2014*) and *L. wollei*'s widespread distribution and high coverage in Florida's springs has become a management concern (*Stevenson et al., 2007*). Therefore, a phage that infects *L. wollei* might serve as a potential biocontrol agent.

## Branch 2: *Pisuviricota* viruses dominate FMA viral diversity

*Pisuviricota* is the most diverse phylum of the *Orthornavirae* kingdom in terms of genome architectures and number of species (*Koonin et al., 2020*). *Pisuviricota* members infect four out of five eukaryotic supergroups, suggesting that viruses from this group evolved prior to the radiation of eukaryotes (*Koonin et al., 2008*). Half of the identified FMA viruses fall within this phylum, with putative members from the orders *Durnavirales*, *Picornavirales*, *Sobelivirales* and *Patatavirales*. These groups are composed of +RNA viruses, with the exception of *Durnavirales* which includes the only double-stranded (ds) RNA viruses within the phylum.

### *Durnavirales* FMA viruses include plant partiti-like viruses and an aquatic picobirna-like virus

*Durnavirales* FMA viral sequences were most similar to members of two distinct families, the *Partitiviridae* and *Picobirnaviridae*. These viruses were preliminarily designated FMA partiti-like viruses 1 through 11 and FMA picorbirna-like virus 1, respectively. Both families are characterized by viruses with segmented dsRNA genomes. Partitiviruses can have more than two segments with two essential genome segments, dsRNA1 and dsRNA2, encoding RdRp and capsid proteins, respectively, whereas picobirnaviruses are bisegmented and each segment encodes a single (RdRp) or two (capsid and hypothetical) proteins. Based on known segment lengths, we identified the near-complete sequence of

**Table 3 Putative plant viruses identified in individual macrophyte species through PCR.**

| Spring (Site)[*] | Species | FMA virus[**] | Top BLASTx match[***] | Identity (%) |
|---|---|---|---|---|
| Blue (1) | Unidentified | levi-like virus 1 | Wenzhou levi-like virus 1 | 42 |
| Blue (2) | Lyngbya wollei | levi-like virus 2[A] | Wenzhou levi-like virus 1 | 72 |
| | Tillandsia usneoides L. | tombus-like virus 6 | Soybean yellow mottle mosaic virus | 37 |
| Iche (1) | Vallisneria americana Michx. | partiti-like virus 3[B] | Rose cryptic virus 1 | 42 |
| Iche (2) | Vallisneria americana Michx. | alphaflexi-like virus 1[C] | Donkey orchid symptomless virus | 38 |
| | | poty-like virus 2 | Potato virus A | 57 |
| | | poty-like virus 3 | Malva vein clearing virus | 64 |
| | | poty-like virus 5 | Pokeweed mosaic virus | 72 |
| | | poty-like virus 6[D] | Turnip mosaic virus | 54 |
| | | poty-like virus 4 | Potato virus B | 33 |
| | | tymo-like virus 2 | Oat blue dwarf virus | 92 |
| | Hydrocotyle umbellate L. | partiti-like virus 3[B] | Rose cryptic virus 1 | 42 |
| | | picorna-like virus 55[E] | Cherry virus Trakiya | 26 |
| Iche (3) | Ludwigia repens J.R. Forst. | partiti-like virus 11 | Melon partitivirus | 65 |
| | | potato virus Y | Potato virus Y | 95 |
| | | poty-like virus 6[D] | Turnip mosaic virus | 54 |
| | Hydrocotyle umbellate L. | partiti-like virus 3[B] | Rose cryptic virus 1 | 42 |
| | | picorna-like virus 55[E] | Cherry virus Trakiya | 26 |
| | Vallisneria americana Michx. | alphaflexi-like virus 1[C] | Donkey orchid symptomless virus | 38 |
| | | picorna-like virus 55[E] | Cherry virus Trakiya | 26 |
| | | sobeli-like virus 1 | Kummerowia striatad enamovirus | 34 |
| Man (1) | Lyngbya wollei | levi-like virus 2[A] | Wenzhou levi-like virus 1 | 72 |
| Man (2) | Lyngbya wollei | levi-like virus 2[A] | Wenzhou levi-like virus 1 | 72 |
| Man (3) | Lyngbya wollei | levi-like virus 2[A] | Wenzhou levi-like virus 1 | 72 |
| Rain (2) | Potamogeton illinoensis Morong | partiti-like virus 4[F] | Rose cryptic virus 1 | 43 |
| | | partiti-like virus 10[G] | Pepper cryptic virus 1 | 43 |
| | Sagittaria kurziana Glück | partiti-like virus 4[F] | Rose cryptic virus 1 | 43 |
| | Utricularia sp | partiti-like virus 10[G] | Pepper cryptic virus 1 | 43 |
| | Ceratophyllum demersum L. | partiti-like virus 10[G] | Pepper cryptic virus 1 | 43 |
| | Najas guadalupensis (Spreng.) Magnus | partiti-like virus 4[F] | Rose cryptic virus 1 | 43 |
| | | partiti-like virus 10[G] | Pepper cryptic virus 1 | 43 |

**Notes:**
[*] Spring sites correspond to sampled locations within each spring (Table 1; Iche, Ichetucknee; Man, Manatee; Rain, Rainbow).
[**] Same superscript letters highlight viruses that were detected in more than one sample.
[***] Accession numbers for BLAST matches are listed in Table S2.

the RdRp-encoding segment of FMA partiti-like viruses 1 through 3 and FMA picobirna-like virus 1 (Fig. 2).

The majority of FMA viruses representing *Durnavirales* (11 out of 12) were most similar to members of the family *Partitiviridae* with matches to partiti-like RdRps or capsids associated with invertebrates ($n = 3$), fungi ($n = 1$), and plants ($n = 7$) (Table S2). A phylogenetic analysis including seven FMA viral RdRps revealed that the majority (five) of these sequences clustered within the genus *Deltapartitivirus*, which is composed of plant

viruses (Fig. 3) (*Vainio et al., 2018*). One of the remaining sequences clustered with the genus *Betapartitivirus*, which includes plant viruses and mycoviruses (*Vainio et al., 2018*), as well as invertebrate-associated partiti-like viruses for which a host has not been determined. Therefore, the majority of FMA partiti-like sequences detected here likely represent novel plant deltapartitiviruses. Remaining sequences may represent invertebrate viruses or mycoviruses, a determination that cannot be made based on sequence similarities and phylogeny alone.

FMA picobirna-like virus 1 is most similar to an arthropod-associated virus, Shahe picobirna-like virus 2 (Table S2). In addition to encoding an RdRp, this putative genome segment encodes a second hypothetical protein with matches to picobirnavirus capsids based on HHpred searches (Fig. 2). This is a novel organization for picobirnaviruses, but some genomes reported from invertebrates, including Shahe picobirna-like virus 2, contain a similar genome organization (*Shi et al., 2016*). Picobirnaviruses were discovered from fecal samples of various vertebrates, mainly mammals, but the picobirnavirus host range remains unclear (*Ghosh & Malik, 2021*). The presence of conserved ribosomal binding sites (RBS) in the genomes of picobirnaviruses reported from vertebrates suggested that these viruses infect bacteria (*Boros et al., 2018*; *Krishnamurthy & Wang, 2018*). The potential host range for picobirnaviruses has been extended to include invertebrates due to their identification in transcriptomes; however, none of the detected viruses were considered transcriptionally active within sampled invertebrates (*Shi et al., 2016*). Additionally, exploration of alternative codons has prompted the possibility that some picobirnaviruses may infect fungi with a lifestyle reminiscent of mitoviruses (*Ghosh & Malik, 2021*) and a scenario where picobirna-like viruses infect unicellular eukaryotes has been noted (*Green et al., 1999*).

A phylogenetic analysis of RdRps found using the standard code confirmed that FMA picobirna-like virus 1 clusters with invertebrate-associated picobirna-like viruses as opposed to clustering with members of the family *Picobirnaviridae*, which includes viruses isolated from vertebrates (Fig. 3). RBS were not detected in FMA picobirna-like virus 1 or any other members of the invertebrate-associated picobirna-like virus clade, which were all retrieved from aquatic environments. This clade also includes a virus discovered from diatoms using a technique that targets intracellular dsRNA, suggesting that the virus infects diatoms (*Urayama, Takaki & Nunoura, 2016*). Although the lack of RBS is not predictive of an eukaryote-infecting virus (*Krishnamurthy & Wang, 2018*), the distinct clade of invertebrate- and diatom-associated picobirna-like viruses with unique genomic features (*i.e.*, lack of RBS and dicistronic segments in some cases) raises the possibility of a picobirnavirus lineage that infects eukaryotes in aquatic environments. More sampling and targeted studies are needed to evaluate this possibility.

### *Picornavirales* FMA viruses are dominated by putative aquatic invertebrate viruses

The order *Picornavirales* encompasses an expansive group of +RNA viruses infecting unicellular organisms, plants, and metazoans and represents the largest order of the phylum *Pisuviricota* (*Koonin et al., 2020*; *Wolf et al., 2018*). FMA viruses from all spring

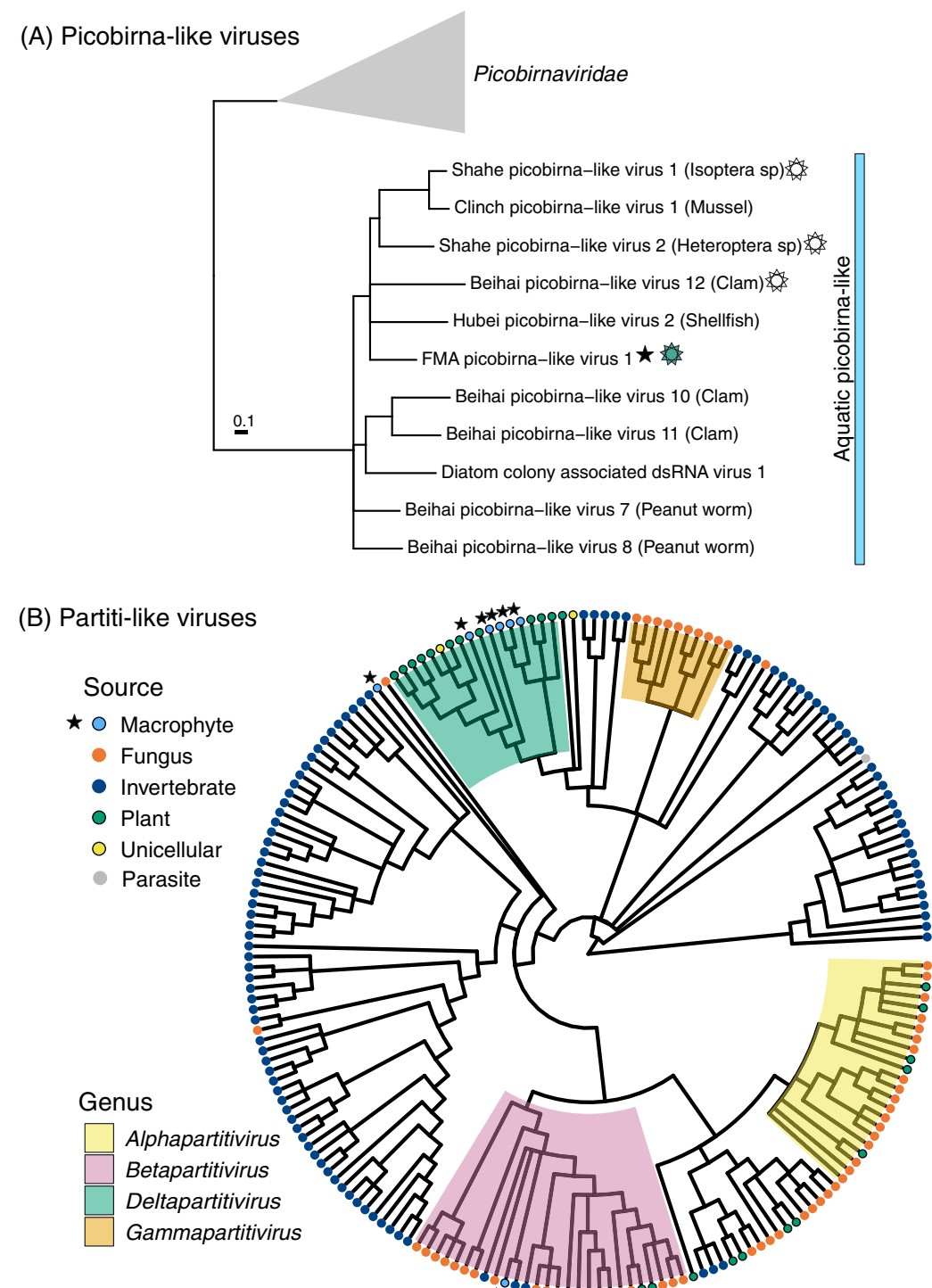

**Figure 3 Midpoint-rooted maximum likelihood phylogenetic trees for members of the order *Durnavirales*, including picobirna-like (A) and partiti-like (B) viruses based on predicted RdRp amino acid sequences.** Freshwater macrophyte associated (FMA) sequences are highlighted with a star. The blue bar on the picobirna-like virus tree highlights sequences that were retrieved from aquatic organisms, which are specified within parenthesis after virus names. The sun symbol points to putative non-segmented genomes in this 'aquatic' picobirna-like clade. Branches with <90% and <70% aLRT support values in picobirna- and partiti-like virus trees, respectively, were collapsed.

sites were dominated by members of this order (Fig. 1A). There are 66 viral contigs matching picorna-like viruses (Table S2), including 16 near-complete genomes that exhibit the typical Helicase-Proteinase-Polymerase domain organization within polyproteins or replication-associated ORFs (Fig. 2) (*Koonin et al., 2008*). These near-complete genomes contained two main genome organizations, monocistronic genomes encoding a polyprotein (*n* = 5) and dicistronic genomes encoding non-structural and structural proteins in separate ORFs (*n* = 11). All five monocistronic genomes have the Helicase-Proteinase-Polymerase-Capsid domain organization reminiscent of some members of the family *Marnaviridae* (*Vlok, Lang & Suttle, 2019a*). Nine of the dicistronic genomes have an organization similar to that of members of the family *Dicistroviridae* with a non-structural ORF followed by a second ORF encoding a dicistro-like capsid. One of the remaining dicistronic genomes does not have a recognizable structural ORF, while the other had a unique organization, with a structural ORF upstream from the non-structural ORF. This novel organization has also been observed in an invertebrate-associated picorna-like virus, Beihai picorna-like virus 105.

The overwhelming majority of FMA picorna-like viral contigs are most closely related to unclassified invertebrate-associated viruses found within this supergroup, with the exception of three contigs that are most similar to picorna-like viruses discovered in seawater (Table S2). Moreover, 61% of FMA picorna-like viral contigs were most similar to invertebrate-associated viruses found within a previously described 'aquatic picorna-like' cluster (*Shi et al., 2016*). This aquatic cluster is most closely related to members of the *Marnaviridae* and includes viruses infecting unicellular photosynthetic organisms (algae and diatoms) and viruses predicted to infect invertebrates given that they appeared to be transcriptionally active within sampled hosts. Phylogenetic analysis including 20 RdRps from FMA picorna-like viral sequences confirmed a high proportion (65%) of contigs clustering within the aquatic picorna-like clade (Fig. S4). Most of the remaining FMA picorna-like viral contigs were most similar to invertebrate-associated members of the *Dicistroviridae*. Therefore, most of the FMA picorna-like viral contigs likely represent novel viruses infecting invertebrates and/or aquatic unicellular eukaryotes.

### *Sobelivirales* FMA viruses potentially represent novel invertebrate viruses

FMA viruses within the order *Sobelivirales*, labelled here FMA sobeli-like viruses, are most similar to 'sobemo-like' viruses (Table S2). Sobemo-like viruses represent a diverse assemblage of invertebrate-associated viruses that are not *bona fide* members of the family *Solemoviridae*, which infect plants (*Shi et al., 2016*; *Wolf et al., 2018*). We identified ten sobeli-like FMA viral contigs, three of which appear to be near-complete genomes based on known genome lengths (Fig. 2). Although these three putative near-complete genomes were most similar to sobemo-like viruses, we did not detect certain features that have been reported from invertebrate sobemo-like viruses (*Shi et al., 2016*). Specifically, we did not detect trypsin-like peptidase domains within any of the RdRp ORFs and only one of the near-complete genomes contained an identifiable capsid-encoding ORF. FMA sobeli-like virus 1 only contained a single ORF encoding RdRp with no recognizable structural domains, which is not typical of sobemo-like viruses or members of the *Solemoviridae*.

FMA sobeli-like virus 2 and 3 have similar genome organization to viruses identified from arthropods including ticks and shrimp, respectively. Given that sobemo-like viruses have been identified in invertebrates, FMA sobemo-like viruses may represent viruses associated with invertebrates that interact with the sampled macrophytes.

### *Patatavirales* FMA viruses include potato virus Y

The order *Patatavirales* is composed of the largest family of RNA plant viruses, namely *Potyviridae* (*Wylie et al., 2017*). We identified six FMA viral contigs with similarities to potyviruses, including a near-complete genome (Fig. 2). Based on limited similarities to known viruses, the viral contigs represent at least three novel potyviruses. However, the FMA potato virus Y (PVY) genome shares 91% genome-wide pairwise identity with a PVY isolate retrieved from potatoes (*Dullemans et al., 2011*). Phylogenetic analysis indicates that the FMA PVY belongs to phylogroup C, which has been identified in tomatoes, peppers, and potatoes and is thought to have diverged in Europe (*Gibbs et al., 2017*) (Fig. 4). In addition to infecting solanaceous crops, PVY is known to infect wild plants, including solanaceous and non-solanaceous weeds, and even ornamentals (*e.g.*, *Kaliciak & Syller, 2009*; *Turina et al., 2006*). Therefore, it is currently unknown if FMA PVY infects freshwater macrophytes and/or reflects agricultural runoff or other terrestrial inputs into the springs.

## Branch 3: *Kitrinoviricota* FMA viruses include putative invertebrate and plant viruses

The phylum *Kitrinoviricota* represents a diverse group of viruses but, in contrast to *Pisuviricota*, only includes +RNA eukaryotic viruses (*Koonin et al., 2020*). *Kitrinoviricota* FMA viruses represent four out of the six orders that currently make up the phylum, including *Hepelivirales*, *Nodamuvirales*, *Tolivirales*, and *Tymovirales* as well as "weiviruses", which have not been classified within an order. Identified viruses within these groups are most similar to plant and invertebrate viruses.

### *Hepelivirales* and "weivirus" FMA viruses are most similar to aquatic viruses

FMA viruses representing members of the *Hepelivirales* and "weiviruses" are most similar to viruses associated with aquatic organisms, mainly invertebrates. The order *Hepelivirales* contains vertebrate, invertebrate, and plant viruses. We identified five FMA hepe-like viruses most similar to viruses associated with aquatic invertebrates (Table S2). One of the contigs, FMA hepe-like virus 1, represents a near-complete genome that was most similar to a putative crab-infecting virus (*Shi et al., 2016*). "Weiviruses" have only been identified from aquatic invertebrates (*Shi et al., 2016*) and seawater (*Wolf et al., 2020*). BLAST searches also revealed a wei-like virus associated with a coral dinoflagellate symbiont, Symbiodinium +RNA virus TR74740 (*Levin et al., 2017*). We detected a single contig similar to "weiviruses", FMA wei-like virus 1, which represents a near-complete genome (Fig. 2). FMA wei-like virus 1 is most similar to Beihai weivirus-like virus 7, a virus detected from an octopus transcriptome (Table S2). FMA wei-like virus 1 contains genomic features similar to those of "weiviruses", including two major ORFs encoding the RdRp and a hypothetical protein.

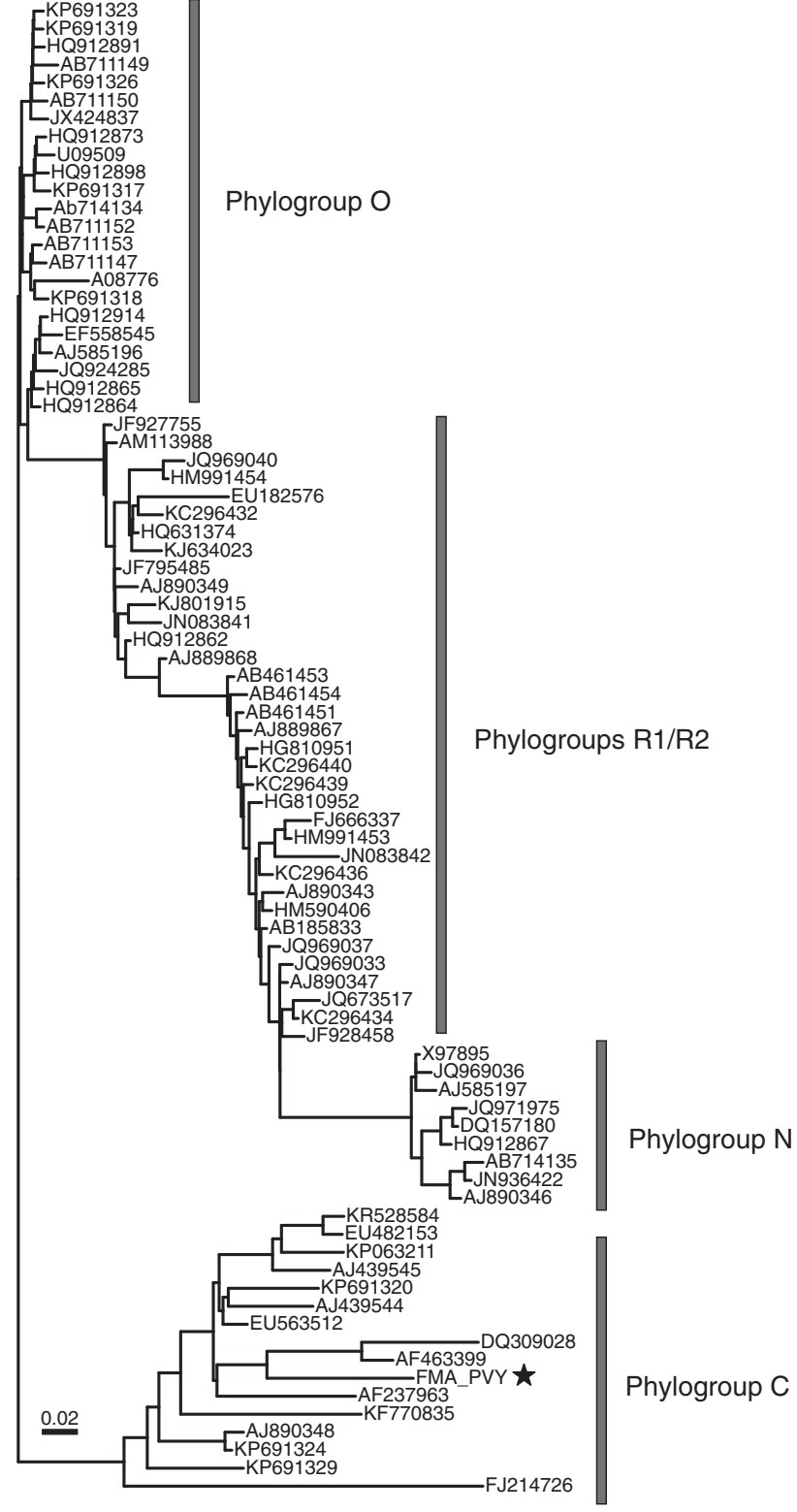

**Figure 4 Midpoint-rooted maximum likelihood phylogenetic tree showing potato virus Y phylogroups (*Gibbs et al., 2017*) based on polyprotein nucleotide sequences.** The freshwater macrophyte associated (FMA) sequence is highlighted with a star. Branches with <70% aLRT support values were collapsed.

### Nodamuvirales and Tolivirales are dominated by putative invertebrate viruses

The majority of FMA viruses representing the orders *Nodamuvirales* and *Tolivirales* are most similar to invertebrate noda-like and tombus-like associated viruses, respectively. We identified four noda-like viral contigs, one of which represented a near complete genome (Fig. 2). Members of the family *Nodaviridae*, which include fish and invertebrate viruses, have bisegmented genomes with the RNA1 segment encoding for the RdRp and RNA2 encoding a capsid protein (*Sahul Hameed et al., 2019*). However, the FMA noda-like virus 1 genome has a dicistronic organization with ORFs encoding for the RdRp and capsid proteins. This organization has also been described from aquatic invertebrate noda-like viruses (*Shi et al., 2016*) suggesting that aquatic noda-like viruses may represent a novel group. Tombusviruses represent a diverse group of plant- and invertebrate-associated viruses (*Wolf et al., 2018*). We identified eight FMA tombus-like viral contigs, including three near-complete genomes (Fig. 2). The three genomes had similar organization to invertebrate tombus-like viruses. Notably, these genomes did not contain ORFs with similarities to movement proteins seen in plant tombusviruses. Therefore, it is likely that the majority of FMA tombus-like viruses are associated with invertebrates.

### Tymovirales FMA viruses include viruses most similar to aquatic and terrestrial plant viruses

The *Tymovirales* is the only order within the phylum *Kitrinoviricota* that is dominated by plant viruses. We detected three FMA tymo-like viral contigs most similar to members from two out of five families within this order, namely *Tymoviridae* and *Alphaflexiviridae*, including a near-complete genome (Fig. 2) (Table S2). The FMA alphaflexi-like virus 1 genome is most similar to donkey orchid symptomless virus, the sole member of the genus *Platypuvirus* within the family *Alphaflexiviridae*. The genus was named after the platypus because the donkey orchid symptomless virus genome encodes proteins from disparate origins, including RdRp and capsid proteins that are related to viruses of the family *Alphaflexiviridae* but a movement protein (MP) that is most similar to that of dianthoviruses in the family *Tombusviridae* (*Wylie, Li & Jones, 2013*). Phylogenetic analysis and genome organization support that FMA alphaflexi-like virus 1 represents a novel member of the *Platypuvirus* genus (Fig. 5). BLAST searches resulted in the detection of a potential third member of this genus based on similarities to the RdRp alone, namely the seagrass-associated virus Cymodea alphaflexivirus 1 (*Bejerman & Debat, 2021*). Therefore, two out of three putative members of the genus *Platypuvirus* are associated with aquatic macrophytes. The remaining two FMA tymo-like viral contigs were most similar to members of the *Tymoviridae*, including one contig, FMA tymo-like virus 2, with high amino acid identity (92%) to oat blue dwarf virus (OBDV). Interestingly, there are other macrophyte viruses in the families *Alphaflexiviridae*, and *Betaflexiviridae*, including turtlegrass virus X (*Van Bogaert et al., 2019*) and Cymodocea nodosa foveavirus 1 (*Bejerman & Debat, 2021*), respectively. The presence of macrophyte viruses in three out of

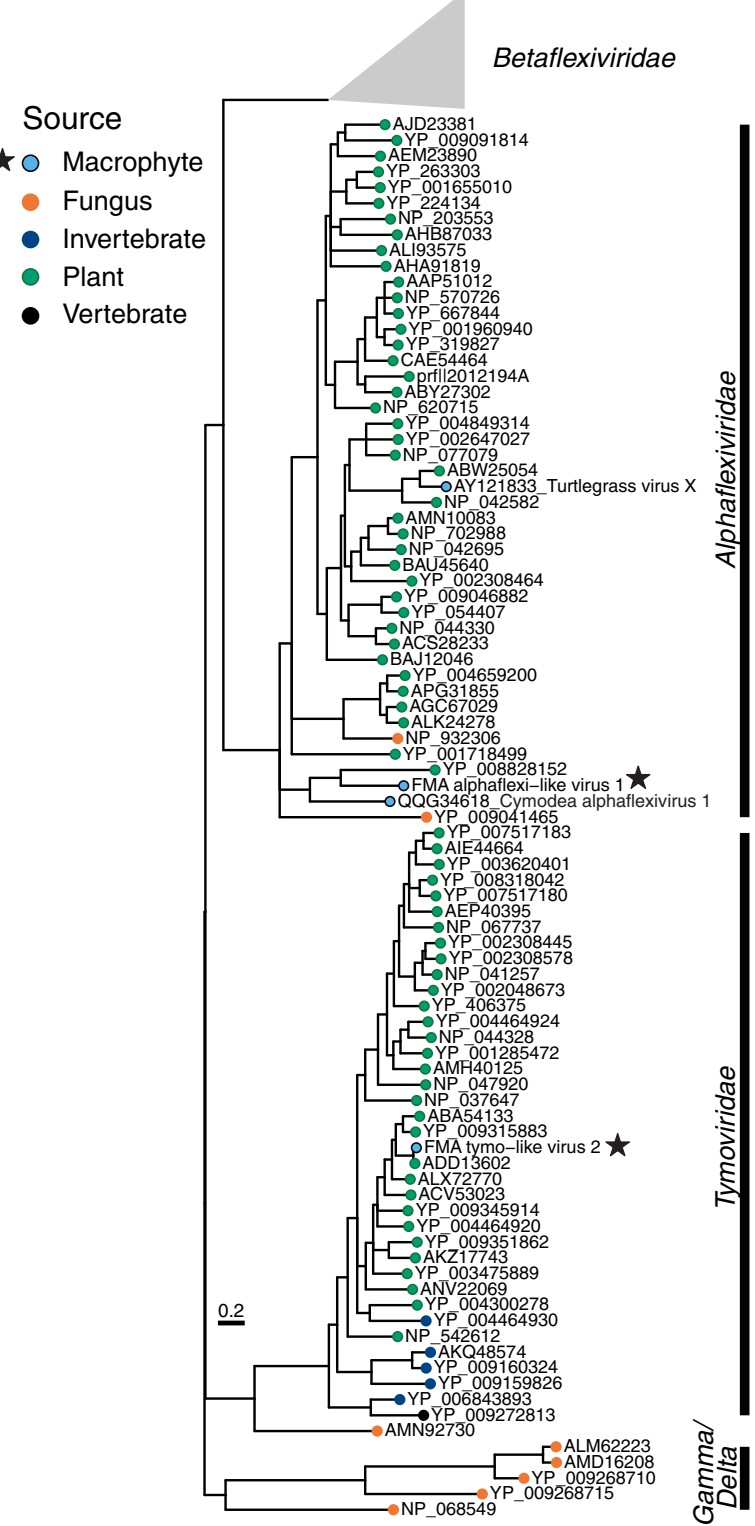

**Figure 5 Midpoint-rooted maximum likelihood phylogenetic trees for members of the order _Tymovirales_ based on predicted RdRp amino acid sequences.** Freshwater macrophyte associated (FMA) sequences are highlighted with a star. Branches with aLRT support values <70% were collapsed. _Gammaflexiviridae_ and _Deltaflexiviridae_ families are abbreviated as 'Gamma' and 'Delta', respectively.

the five families within the *Tymovirales* indicates that viruses from this order might thrive in both terrestrial and aquatic plants.

## Branch 4: *Duplornaviricota* FMA viruses include putative novel mycoviruses- and diatom-infecting viruses

The phylum *Duplornaviricota* includes the majority of known eukaryotic dsRNA viruses, which are distributed among three orders, namely *Reovirales*, *Ghabrivirales*, and *Mindivirales*. We identified 16 FMA viral contigs most similar to members of the order *Ghabrivirales* (Table S2), including two near-complete genomes or segments (Fig. 2). Members of the *Ghabrivirales* include viruses with a diverse array of genome lengths and arrangements, which infect protists, fungi, plants, and metazoans (*Wolf et al., 2018*). Nine of the FMA ghabri-like viral contigs were most similar to mycoviruses from the family *Megabirnaviridae*. The remaining ghabri-like viral contigs were most similar to unclassified viruses associated with diatoms ($n = 6$) and mosquitoes ($n = 1$). FMA ghabri-like viral contigs most similar to diatom-associated viruses had matches to diatom colony associated dsRNA virus 16, which had high coverage in RNA libraries targeting diatom intracellular dsRNA and likely infects diatoms (*Urayama, Takaki & Nunoura, 2016*). The two FMA ghabri-like near complete genomes represent either a single segment encoding RdRp and capsid proteins, similar to segment 1 of *Megabirnaviridae* (*Sato et al., 2019*), or 'minimal' dsRNA genomes, similar to members of the *Totiviridae* (*Wolf et al., 2018*). Phylogenetic analysis indicates that the four FMA ghabri-like viral RdRps cluster closely with segmented viruses from the *Megabirnaviridae* and invertebrate-associated toti-like viruses (Fig. S5). Altogether, these analyses suggest that the majority of FMA ghabri-like viral contigs likely represent novel mycoviruses and/or diatom-associated viruses.

## Branch 5: *Negarnaviricota* FMA viruses potentially represent protist and invertebrate viruses

The phylum *Negarnaviricota* includes the vast majority of negative-sense RNA viruses currently classified by the ICTV. We identified FMA viruses most similar to viruses from four out of seven orders within this phylum, including *Muvirales*, *Goujianvirales*, *Bunyavirales*, and *Jingchuvirales*. Members of the *Muvirales* and *Goujianvirales* were discovered in invertebrates and nematodes (*Shi et al., 2016*) and are classified within the families *Qinviridae* and *Yueviridae*, respectively. Sequences similar to qinviruses and yueviruses have also been reported from protists (*Chiapello et al., 2020*) and soil samples (*Starr et al., 2019*). Additionally, BLAST searches revealed qin- and yue-like viral proteins from protozoan (*Brachionus plicatilis*, accession no. RNA03874), algal (*Chara braunii*, accession no. GBG68844) and plant (*Vigna unguiculata*, accession no. QCE01079) genome projects, suggesting that these viruses may be associated with a diversity of organisms including autotrophs. We identified four FMA yue-like viral contigs, including a near-complete genome (Fig. 2), and three FMA qin-like viral contigs (Table S2). The FMA yue-like viral contigs were most similar to yueviruses associated with freshwater invertebrates, whereas FMA qin-like viruses had matches to nematode and soil viruses, as

well as proteins associated with algae and plants. The FMA yue-like virus 1 contig may represent a single monocistronic segment encoding the RdRp, given that yueviruses are bisegmented (*Shi et al., 2016*). Phylogenetic analysis indicates that detected FMA yue- and qin-like viruses cluster with viruses associated with protists, including oomycetes and rotifers (Fig. S6).

*Bunyavirales* and *Jingchuvirales* are two large orders within the *Negarnaviricota* that contain viruses with segmented and mainly unsegmented genomes, respectively. We identified two FMA bunya-like viral contigs. FMA bunya-like virus 1 represents a near-complete segment encoding the RdRp (L segment) (Fig. 2), which is most closely related to a mycovirus, Rhizoctonia solani bunya/phlebo-like virus 1 (Table S2). Phylogenetic analysis indicates that FMA bunya-like viral sequences cluster closest to oomycete viruses (*Chiapello et al., 2020*) and Rhizoctonia solani bunya/phlebo-like virus 1 (*Picarelli et al., 2019*), in a clade composed of viral genomes for which structural proteins have not been reported (Fig. S7). This clade also includes viruses associated with cestodes (Schistocephalus solidus bunya-like virus 1) (*Hahn et al., 2020*) and invertebrates (Beihai barnacle virus 5 and Barns Ness serrated wrack bunya/phlebo-like virus 1) (*Shi et al., 2016*; *Waldron, Stone & Obbard, 2018*). Whereas FMA bunya-like viruses are most closely related to mycoviruses and oomycete viruses, FMA jingchu-like viruses are most similar to invertebrate-associated viruses. We identified four FMA jingchu-like viral contigs, including a near-complete genome (Fig. 2). The FMA jingchu-like virus 1 is most similar to viruses identified in snakes and freshwater shrimp based on BLAST searches using ORFs encoding the RdRp and glycoprotein (Table S2). Phylogenetic analysis did not provide further insight since the FMA jingchu-like virus 1 RdRp sequence clustered just outside a clade including snake and shrimp-associated viruses (Fig. S8). However, the nucleoprotein encoding ORF is most similar to insect-associated viruses. BLAST searches using the FMA jingchu-like virus 1 genome as a query resulted in top matches to a glycoprotein detected in the transcriptome from a freshwater amphipod (accession number XP_018024392). Therefore, we suspect that FMA jingchu-like viruses represent invertebrate-associated viruses rather than vertebrate-infecting viruses.

## Putative FMA plant viruses are widespread in vascular aquatic vegetation

Given the limited information regarding viral infection in freshwater aquatic vegetation, we used PCR to identify which macrophyte species contained putative FMA plant viruses. Based on BLAST matches and/or phylogenetic analysis we identified 15 distinct FMA viral contigs potentially representing plant viruses (Table 3). All of the putative FMA plant viruses were detected in perennial, mainly vascular, plants and the majority (10 out 15) were limited to a single macrophyte species, *Vallisneria americana* Michx. Poty-like (40%) and partiti-like (27%) viruses most similar to members of the *Potyviridae* and *Partitiviridae* families, respectively, dominated the diversity of FMA viruses representing plant viruses. However, poty-like viruses were only identified in two macrophyte species sampled in Ichetucknee, namely *V. americana* and *Ludwigia repens* J.R. Forst., whereas cryptic partiti-like viruses were detected in eight species collected throughout Ichetucknee

and Rainbow spring sites. This observation extends to individual FMA viruses given that the most prevalent FMA virus was FMA partiti-like virus 10, which was detected in four macrophyte species. Therefore, partiti-like viruses most similar to persistent cryptic viruses that cause asymptomatic infections (*Vainio et al., 2018*) may be more widespread in spring macrophytes than poty-like viruses.

The putative plant viruses we identified in freshwater macrophytes reflect what has been observed in natural terrestrial ecosystems, where there are an abundance of viral groups with persistent lifestyles (*Prendeville et al., 2012*; *Roossinck, 2015*). At the time of macrophyte sample collection there were no evident signs of symptomatic viral infection, and all macrophytes where putative plant viruses were detected are considered perennials. Persistent viruses, which establish long-term infections, are expected to spread in their host plant without causing critical damage and such a lifestyle may be more easily maintained in perennial hosts (*Shates et al., 2019*; *Takahashi et al., 2019*). Although persistent lifestyles are typically associated with viruses with dsRNA genomes (*Roossinck, 2010*), a diversity of +RNA viruses also establish persistent infections (*Takahashi et al., 2019*). Future research should address the role of persistent viral infection in the ecology of aquatic vegetation given that these interactions may have important implications for the host plant, including potential beneficial outcomes (*Roossinck, 2011*; *Takahashi et al., 2019*).

The majority of putative FMA plant viruses had low amino acid level identities to known viruses, indicating their genetic novelty (Table 3). However, FMA tymo-like virus 2 and FMA PVY had high similarities (>85% nucleotide identity) to oat blue dwarf virus (OBDV) and potato virus Y (PVY), respectively. OBDV infects grasses of economic interest, such as oat and maize (*Edwards & Weiland, 2009*; *Edwards & Weiland, 2010*), and PVY is a major agricultural pathogen (*Torrance & Talianksy, 2020*). Although OBDV and PVY have broad host ranges and infect weeds that may serve as viral reservoirs (*Cervantes & Alvarez, 2011*; *Westdal, 2011*), to our knowledge, this is the first study to detect these viruses in macrophytes. Some PVY strains are transmissible in water suggesting that water-mediated infection is plausible (*Mehle et al., 2014*). However, OBDV is a phloem-limited virus that is transmitted by insect vectors and the mechanism of its introduction to submerged aquatic vegetation is less clear. Regardless, our findings highlight that macrophytes may harbor terrestrial plant viruses, a possibility that was noted over 70 years ago when researchers observed lesions in terrestrial indicator plants inoculated with homogenates from aquatic plants (*MacClement & Richards, 1956*). However, the identities of the viruses causing symptoms in terrestrial indicator plants were not determined in that study. Future work will examine the distribution of PVY and OBDV in freshwater springs, if their presence in macrophytes is related to agricultural runoff, and if these viruses can replicate and be transmitted among macrophytes.

*Vallisneria americana* Michx was the macrophyte species with the highest diversity of putative plant viruses resulting in the detection of ten viruses, including OBDV. Half of the viruses detected in this macrophyte represent poty-like viruses. The global distribution, wide host range (including cultivated and wild vegetation), and fixed hypervariable genomic regions of members of the *Potyviridae* point to their adaptability to new hosts and environments (*Nigam et al., 2019*; *Wylie et al., 2017*). The detection of a diversity of

poty-like viruses in *V. americana* indicate that potyviruses may also thrive in freshwater vegetation. Notably, *V. americana* is a submerged monocotyledonous plant and freshwater member of the Alismatales, the only order also encompassing marine flowering plants (*i.e.*, angiosperms), namely seagrasses. Poty-like viruses have not been reported yet from marine angiosperms; however, there are reports of brown macroalgae containing flexuous virus particles reminiscent of potyviruses that reacted to potyvirus monoclonal antibodies (*Easton, Lewis & Pearson, 1997*).

The findings presented here, together with reports from seagrasses (*Bejerman & Debat, 2021*; *Van Bogaert et al., 2019*), suggest that some viruses identified in freshwater and marine angiosperms fall within the known diversity of terrestrial plant viruses. This was not expected considering that the core Alismatales, an order mainly composed of plants that have a completely submerged seedling phase, evolved over 120 million years ago (*Du & Wang, 2016*; *Givnish et al., 2018*). Aquatic angiosperms are generally thought to have evolved from terrestrial plants (*Les, Cleland & Waycott, 1997*; *Papenbrock, 2012*). However, more recent analyses suggest an alternative scenario where some angiosperm lineages, including the Alismatales, originated and dispersed in aquatic environments (*Du & Wang, 2016*; *Givnish et al., 2018*; *Gomez et al., 2015*). Our findings suggest that despite the divergence between aquatic and terrestrial angiosperms, there may be close evolutionary relationships among their viruses. Moreover, the discovery of a diversity of RNA viruses in microalgae (Chlorophyta and Chlorarachniophyceae) suggest that there may be more complex and closer evolutionary relationships among viruses infecting aquatic phototrophs and land plants than previously thought (*Charon et al., 2020*). More sampling of unexplored aquatic phototrophs, including macrophytes, is needed to evaluate how virome composition in aquatic primary producers relates to changes in plant evolution (*Mushegian, Shipunov & Elena, 2016*) and the potential ecological impacts of viral infection in macrophytes.

## CONCLUSIONS

Here we described viral diversity associated with macrophytes from freshwater springs, further expanding the known RNA viral diversity associated with aquatic phototrophs. FMA viruses include viruses associated with a diversity of organisms that are presumed to be part of macrophyte holobionts as well as organisms that directly interact with macrophytes (*e.g.*, invertebrate herbivores). Although macrophytes play a vital role in aquatic ecosystems, viral infection may have escaped detection due to persistent and asymptomatic infections that go unnoticed. Despite the lack of obvious symptomatic disease, persistent viral infections play important roles in terrestrial plant ecology (*Lefeuvre et al., 2019*; *Roossinck, 2015*; *Takahashi et al., 2019*) and are likely to affect macrophytes. Known terrestrial plant pathogens identified in submerged aquatic vegetation highlight a potential terrestrial-aquatic continuum for plant viruses. This is important since agricultural runoff is considered one of the major nonpoint pollution sources impacting freshwater systems (*Xia et al., 2020*) and it is likely that abiotic and biotic farmland components, such as viruses, reach aquatic habitats. Although there have been concerns about plant viral pathogens in freshwater sources used for crop irrigation (*Hong, 2017*;

*Mehle & Ravnikar, 2012*; *Rosario et al., 2009*), the potential effects of terrestrial plant viruses in aquatic vegetation have not been explored. The springs act as 'natural flowing water chemostats' with unparalleled temporal stability in physicochemical parameters that render them suitable for studying autotroph homeostasis (*Nifong, Cohen & Cropper, 2014*). The genetic data gathered here can be used to design molecular assays to investigate virus-macrophyte interactions in these natural freshwater laboratories and further investigate plant virus evolution by confirming virus-host associations.

## ACKNOWLEDGEMENTS

The authors would like to thank the Florida Department of Environmental Protection Division of Recreation and Parks and the Florida Park Service for their assistance in obtaining sampling permits and access to the springs. The authors also acknowledge Kema Malki, Kaitlin Mettel, Robin Jung and William E. Landry for all their help during field sample collection and sample processing.

### Funding

This project was funded by grant DEB-1555854 from the National Science Foundation to Mya Breitbart and a grant to Noémi Van Bogaert from the Belgian American Educational Foundation (BAEF). The funders had no role in study design, data collection and analysis, decision to publish, or preparation of the manuscript.

### Grant Disclosures

The following grant information was disclosed by the authors:
National Science Foundation: DEB-1555854.
Belgian American Educational Foundation (BAEF).

### Competing Interests

Mya Breitbart is an Academic Editor for PeerJ.

### Author Contributions

- Karyna Rosario conceived and designed the experiments, performed the experiments, analyzed the data, prepared figures and/or tables, authored or reviewed drafts of the article, and approved the final draft.
- Noémi Van Bogaert conceived and designed the experiments, performed the experiments, authored or reviewed drafts of the article, and approved the final draft.
- Natalia B. López-Figueroa performed the experiments, authored or reviewed drafts of the article, and approved the final draft.
- Haris Paliogiannis performed the experiments, authored or reviewed drafts of the article, and approved the final draft.
- Mason Kerr performed the experiments, authored or reviewed drafts of the article, and approved the final draft.

- Mya Breitbart conceived and designed the experiments, authored or reviewed drafts of the article, and approved the final draft.

## Field Study Permissions

The following information was supplied relating to field study approvals (*i.e.*, approving body and any reference numbers):

Field samples were collected from freshwater springs in Florida, USA in accordance with a permit from the Florida Department of Environmental Protection (permit # 06011710).

## DNA Deposition

The following information was supplied regarding the deposition of DNA sequences:

Near-complete RNA viral genomes or segments are available at GenBank: ON125107 to ON125143.

## Data Availability

Next generation sequencing data are available at the Sequence Read Archive (SRA) database under BioProject PRJNA826216 (BioSample accession numbers SAMN27553344 through SAMN27553353).

## Supplemental Information

Supplemental information for this article can be found online at http://dx.doi.org/10.7717/peerj.13875#supplemental-information.

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
