# Peer review of "Freshwater macrophytes harbor viruses representing all five major phyla of the RNA viral kingdom Orthornavirae"

_PeerJ, doi:10.7717/peerj.13875_

## Round 0.1 · original submission · Minor Revisions

Dear Dr. Rosaria and colleagues:

Thanks for submitting your manuscript to PeerJ. I have now received three independent reviews of your work, and as you will see, the reviewers raised some minor concerns about the research. Despite this, these reviewers are optimistic about your work and the potential impact it will lend to research on macrophyte viromes. Thus, I encourage you to revise your manuscript, accordingly, taking into account all of the concerns raised by the reviewers.

Please remember to submit all novel sequences to NCBI.

I look forward to seeing your revision, and thanks again for submitting your work to PeerJ.

Good luck with your revision,

Best,

-joe

·

Basic reporting

This manuscript is clearly and unambiguously written. The literature references along with background and context are fully provided. The article is professional in structure and content with the results fully supporting the goal of filling a significant gap of knowledge.

The article should be published as is. No further editing is needed.

Experimental design

This is one of the most complete descriptions of experimental design I have seen. The information is relevant and needed for the structure and analyses of the paper. No edits or additions are needed.

Validity of the findings

This is one of the most significant plant virology papers written in many years. No edits or additions are needed.

Additional comments

A tour de force. The paper will significantly re-calibrate the field of plant virology and become the foundational chapter of aquatic plant virology. Stunning.

·

Basic reporting

No comment

Experimental design

No comment

Validity of the findings

No comment

Additional comments

I have read the manuscript numerous times and not found a single concern. The manuscript is well written and suitable for immediate publication. The authors have provided the necessary details and documentation typical of any viral discovery paper and have performed the analyses one would expect to see for this type of work. I applaud and thank the authors for doing such a wonderful job.

Reviewer 3 ·

Basic reporting

This is a very well-written paper with a clear structure, organized reporting, professional English, and sound questions/answers/results.

Experimental design

The article is within scope and all investigations are performed rigorously and according to the current state of the art. Results are meaningful and important and methods are described in sufficient detail.

Validity of the findings

The findings are highly relevant for further definition of the virosphere and the accomplished dataset is unique and comprehensive. All data are provided, but sequences need to be submitted to GenBank.

Additional comments

My main concerns are primarily editorial in nature:

- L22: isn't research included in -ology, i.e., isn't "virology research" redundant?
- L25 and elsewhere: I thought NGS is now a discouraged term and HTS is instead recommended?
- L45 and probably elsewhere: add PMID: 35638822?
- L56, 377: producers of what?
- L100: how are they unique (other than any habitat being unique)?
- L256 and elsewhere: change "size" [of contigs/sequences/nucleic acids] to "length"
- L270 and elsewhere: the correct term (per IUBMB/IUPAC) is "RNA-directed RNA polymerase"
- L338-339: a bit confusing - contigs don't have amino acids
- L340 and elsewhere: please replace "named" with "designated" or "labeled". Naming should be reserved for classifiable viruses (i.e., coding-complete genomes). Case in point is L463 etc.: these “names” are not sustainable as new taxa may have to be established next to established “partiti” taxa, which will make these names confusing. Thus, “here preliminary designated” would be preferrable throughout
- L340 and elsewhere: please replace "level" with "rank" in context of taxa
- L341-343 and elsewhere: you could simplify the writing. Instead of writing "members of the order _Picornavirales_", "picornavirals" would be just fine
- L361-362 and elsewhere: "diatom-like viruses" and "algal viruses" doesn't make sense. Diatoms are nothing like viruses :)
- L390-394 and elsewhere: this needs to get fixed: the family _Leviviridae_ and order _Levivirales_ have been abolished; instead there is now a class _Leviviricetes_ with several families and hundreds of genera. Please see and also cite PMID: 34747690
- L406 and possible elsewhere: virus names are not to be capitalized except if a virus name component is a proper noun or single letter (i.e., should be "apple")
- L410 and elsewhere: it would be very helpful to the reader it the vernacular designations for the members of taxa had taxon-specific suffixes so the discussed rank can be deduced. In particular, next to the -virals suggested above, family members should be referred to as -virids (here: botourmiavirids infection...)
- L420 and elsewhere: please add missing Pīnyīn marks to Chinese virus name components: Wēnzhōu, Shāhé, Běihǎi
- L460 and similar headers: if the headers are in italics, then taxon names should be not italicized within. Please also change to vernacular names/adjectives in these headers, i.e., “Pataviral FMA viruses…” etc.
- L585 and elsewhere: please place “weiviruses” in quotation marks as it is not an official term
- L629: based on line 628 the genus was not named ‘platypus’ [s is missing]. Rephrase
- L668: Based on PMID: 35389782 (“Arctiviricota”) and PMID: 35082445 (“epsilon-“, “zetaviruses” in addition to numerous deltaviruses etc.) I would be cautious with this statement
- L687: _Jingchuvirales_, not _Jungchivirales_. And I have doubts about that. _Mononegavirales_ appears much bigger to me
- L717 and elsewhere: Please add the mandated authorities to plant species names (i.e., here should be “_Vallisneria_ americana Michx.”). Also please ensure not to mix taxa (concepts) and assigned members (things/organisms), i.e., here should be “namely eelgrass (_Vallisneria_ americana Michx.)”.
- L724: “detected in four macrophytes” or “detected in macrophytes of four species”
- L744-747: where the abbreviations OBDV and PVY introduced earlier?
- Table 1: In column 3, the information should be reverse. Title should be “Collected organism (species)”, content should be akin to “eelgrass (_Vallisneria_ americana Michx.)”. Lynbia wollei should be italicized. Copy this to column 2 in Table 3 (a species is not a specimen)
- Figure 1A: complete “Permutotetra”

---

## Round 0.2 · accepted · Accept

Dear Dr. Rosario and colleagues:

Thanks for revising your manuscript based on the concerns raised by the reviewers. I now believe that your manuscript is suitable for publication. Congratulations! I look forward to seeing this work in print, and I anticipate it being an important resource for groups studying macrophyte viromes. Thanks again for choosing PeerJ to publish such important work.

Best,

-joe